# Comparison of the Processability and Influence on the Microstructure of Different Starting Powder Blends for Laser Powder Bed Fusion of a Fe$_{3.5}$Si$_{1.5}$C Alloy

Anna Luise Strauch [1,*], Volker Uhlenwinkel [1,2], Matthias Steinbacher [1,3], Felix Großwendt [4], Arne Röttger [5], Abootorab Baqerzadeh Chehreh [6], Frank Walther [6] and Rainer Fechte-Heinen [1,3]

[1] Leibniz Institute for Materials Engineering, Badgasteiner Straße 3, 28359 Bremen, Germany; uhl@iwt.uni-bremen.de (V.U.); steinbacher@iwt-bremen.de (M.S.); fechte@iwt-bremen.de (R.F.-H.)
[2] Department of Production Engineering, Particle and Process Engineering, University of Bremen, Badgasteiner Str. 1–3, 28359 Bremen, Germany
[3] MAPEX Center for Materials and Processes, University of Bremen, 28359 Bremen, Germany
[4] Chair of Materials Technology, Ruhr-University Bochum, Universitaetsstr. 150, 44780 Bochum, Germany; felix.grosswendt@ruhr-uni-bochum.de
[5] Chair of New Manufacturing Technologies and Materials, University of Wuppertal, Bahnhofstr. 15, 42651 Solingen, Germany; roettger.fuw@uni-wuppertal.de
[6] Department of Materials Test Engineering, TU Dortmund University, Baroper Str. 303, 44227 Dortmund, Germany; abootorab.chehreh@tu-dortmund.de (A.B.C.); frank.walther@tu-dortmund.de (F.W.)
[*] Correspondence: strauch@iwt-bremen.de

**Abstract:** This paper examines different blends of starting materials for alloy development in the laser powder bed fusion (LPBF) process. By using blends of individual elemental, ferroalloy and carbide powders instead of a pre-alloyed gas-atomized starting powder, elaborate gas-atomization processes for the production of individual starting powders with varying alloy compositions can be omitted. In this work the model alloy Fe$_{3.5}$Si$_{1.5}$C is produced by LPBF from different blends of pure elemental, binary and ternary powders. Three powder blends were processed. The base material for all powder blends is a commercial gas-atomized Fe powder. In the first blend this Fe powder is admixed with SiC, in the second with the ternary raw alloy FeSiC and in the third with FeSi and FeC. After characterizing the powder properties and performing LPBF parameter studies for each powder blend, the microstructures and the mechanical properties of the LPBF-manufactured samples were analyzed. Therefore, investigations were carried out by scanning electron microscopy, wave length dispersive x-ray spectroscopy and micro hardness testing. It was shown that the admixed SiC dissolves completely during LPBF. But the obtained microstructure consisting of bainite, martensite, ferrite and retained austenite is inhomogeneous. The use of the lower melting ferroalloys FeSi and FeC as well as the ternary ferroalloy FeSiC leads to an increased chemical homogeneity after LPBF-processing. However, the particle size of the used components plays a decisive role for the dissolution behavior in LPBF.

**Keywords:** additive manufacturing; laser powder bed fusion; powder blending; steel powder

## 1. Introduction

Today, individual metal powder alloys for Laser Additive Manufacturing (LAM) processes such as Laser Powder Bed Fusion (LPBF) are difficult to achieve due to the complex LPBF processability and the complicated atomization process. Therefore, only few alloys can be found in the market. However, laser additive manufacturing is an emerging technology offering a variety of different process-related advantages compared to subtractive manufacturing.

LPBF uses a powder feedstock, which is deposited on a building platform and selectively fused by a computer-controlled laser beam under an inert gas atmosphere [1,2].

After applying a new layer of powder, the process is repeated until the part is completely built up in the powder bed [1]. Thereby, LPBF is characterized by a high amount of geometric freedom. Accordingly, undercuts or inner cavities can be realized which cannot be produced by conventional manufacturing routes.

However, nowadays only a few Fe alloys are commercially available for LPBF processing. These Fe alloys include the austenitic stainless steel 316L, but wear-resistant alloys are hardly available [3]. The rapid solidification rate occurring in the layer-wise material deposition in LPBF promotes crack formation in many conventional grades of carbon-martensitic hardenable tool steels or other wear-resistant Fe alloys like hard alloys or white cast irons. To counteract the crack formation in such Fe alloys during the LPBF process, new alloy compositions must be derived [4–6]. On the other hand, a reason for the limited range of available Fe based starting powders is the use of the gas-atomization process, which generates a spherical particle shape but produces only a portion of the generated powder in the required size range. This technique is applied to ensure sufficient flowability and chemical homogeneity.

Consequently, this work aims to support the development of wear resistant Fe alloys specially adapted for LPBF by investigation the use of powder blends to omit elaborate gas-atomization processes. Individual alloys can be flexibly blended from a small stock of raw material powders without the need to produce and store various specific pre-alloyed gas-atomized steel powders. Significant flexibility in alloy development can be gained [7,8]. Simultaneously, increasing speed for development of new parts based on optimal alloy design, manufacturing process, and heat treatment can be achieved [9].

Mixing, blending and in-situ alloying for the production of adapted alloys have already been presented for LAM for different materials [9], such as high entropy alloys (HEAs) [8,10–12], aluminum alloys [13–18], titanium [19–22] or stainless steels [23], but also for different LAM technologies [15,24,25]. Studies on wear-resistant Fe-based alloys are rarely found [9,26]. In the respective literature, elemental powders are mostly used in blends and in-situ alloys, but ferroalloys are rarely used. These have a lower liquidus temperature than many pure elements. In previous experiments, it was shown that the ferroalloys FeCrC, FeSi, and FeTi, for example, can be homogeneously processed by LPBF, whereas FeW and FeMo tend to stay unmolten [26]. The reason for this is probably the interaction of the particle size and the liquidus temperatures of the respective alloys. The energy input into the particles was possibly too low during the LPBF process, so that the melting of the particles remained incomplete.

It is known that even some elemental powders, such as Ni [27], which have a similar melting temperature compared to Fe, can be dissolved during LAM. In contrast, higher melting elements, such as Cr or W, can hardly be dissolved in the melt pool [12,17,23,26]. The incomplete melting of raw material particles results in non-uniform chemical compositions as well as in high porosity [15,20,25,28]. The influence of the process parameters used, such as the volumetric energy density (VED) and the size of the melt pool [8,10,23], as well as the melting temperature of the individual powder alloys, must be considered when processing powder blends by LPBF [9,22]. Yet, the homogeneity of the produced components can be improved by optimized process parameters or post-process heat treatments [20,29,30]. Further investigations are necessary in order to predict which powder blends are suitable to achieve homogeneous materials after LPBF processing of Fe-based powder blends.

In order to support alloy development for a wide variety of Fe-based alloys, the focus of this study will be limited to two major alloying elements used in wear-resistant Fe alloys: silicon (Si) and carbon (C). The selected model alloy with a C content of 1.5 wt. % was selected with wear resistant ledeburitic cold work steels in mind. Such materials are used for cutting tools which are exposed to high abrasive wear. The model alloy is intended to show how high C contents can be introduced into the microstructure by powder blending and how the added C sources locally affect the microstructure formation. Using Si as the only substitutional alloying element, the phase formation derived during

cooling is simplified because the formation of Fe carbide is suppressed. By decreasing the number of alloying elements, the model system is simplified and the focus lays on the influence of the different starting alloys. It is intended to show how the use of different raw materials affects the powder blending, the optimal processing parameters, the chemical homogeneity, the phases and the microstructure formation, and the local hardness of the LPBF-produced materials.

## 2. Materials and Methods

In the following, the model alloy composition to be obtained after LPBF processing will be referred to as target alloy and the raw materials used for blending are defined as raw alloys. The target alloy is of the chemical composition $FeSi_{3.5}C_{1.5}$ comprising the two alloying elements: Si and C, commonly used in wear resistant Fe alloys and cast irons. The raw alloys used are Fe, SiC, $Fe_7Si$, $Fe_{4.3}C$ and $Fe_{5.5}Si_{2.35}C$. Further information on the used powders follows in Section 3.2. Three different powder blends are made from these raw alloys to produce the target alloy. By using these powder blends, the impact of the elemental, binary and ternary raw alloys on the manufactured samples is investigated.

In some publications, this method of producing alloys is referred to as in-situ alloying [9,10,23,28,31], mixing [13,14,18,21,24,32] or blending [8,11,15,28,33], among other terms. In this work, the term "powder blending" is used.

### 2.1. Gas Atomization and Powder Blending

The conventional way to produce an alloy for LPBF is to gas-atomize a pre-alloyed melt [34]. Here, a gas-atomized pure Fe powder from Nanoval GmbH & Co. KG (Berlin, Germany) was blended with the binary and ternary raw alloys. The raw alloys $Fe_7Si$, $Fe_{4.3}C$ and $Fe_{5.5}Si_{2.35}C$ were produced by means of nitrogen gas-atomization in a close-coupled atomizer (AU 1000 Prototype, Indutherm, Walzbachtal, Germany) to obtain spherical particles. In contrast, the SiC powder was produced by milling.

After the atomization of the raw alloys, the fine fraction < 20 μm was removed by air classification (Multiprocess Airclassifier, Hosokawa Alpine AG, Augsburg, Germany) and then sieved by an air jet sieve (Air Jet Sieve e200LS, Hosokawa Alpine AG, Augsburg, Germany) to obtain a fraction of 20–63 μm. A 3D shaker mixer (Turbula® T2F, WAB Group, Muttenz, Switzerland) was used for blending of the powders. The powders were weighed in the specified mass percentages and mixed for 10 min at 98.4 rpm to obtain the three different powder blends.

### 2.2. Powder Characterization

Due to the fact that the powder properties influence the behavior in the LPBF process and thus the additively manufactured microstructure [31], the different powder blends are investigated with regard to particle size distribution, morphology and flowability. This can be helpful in interpreting the generated microstructure and the associated properties.

The particle size distributions were analyzed with a diffraction spectrometer (Mastersizer 2000, Malvern Panalytical Ltd., Malvern, UK). A particle refractive index of 2.86 was used for all Fe-based materials and an index of 3.5 for the SiC. The flowability is measured by the Hall flow test according to DIN EN ISO 4490 with a 2.5 mm Nozzle and the tap density according to DIN EN ISO 3953 to determine the Hausner Ratio and the evaluation was conducted following Carr et al. [35].

The C-content is measured with a CS744 (LECO Corporation, St. Joseph, MI, USA) using hot gas extraction. For this method, the mean value was calculated from five measurements per blend. To measure the Fe-content, atomic absorption spectrometry was used and the Si-content was analyzed gravimetrically. For SiC and Fe4.3C, only the C content was determined and the remainder was assumed to be Fe or Si content. The solidus and liquidus temperatures were calculated by ThermoCalc™ using Database TCFE7 (Version 2020a). Various values can be found in the literature for the solidus and liquidus temperatures of SiC Here, the reference to the values of Franke et al. is to be given [36].

### 2.3. LPBF Parameters

The cuboid samples with an edge length of $4 \times 4 \times 10$ mm were built with 1 mm support structure on a cylindrical building platform with a diameter of 55 mm in an argon atmosphere by means of LPBF (AconityMINI, Aconity GmbH, Herzogenrath, Germany). For the laser scanning strategy, a checkerboard strategy was used, with an island size of $1 \times 1$ mm, which are rotated 90° to each other. Further process parameters can be found in Table 1. In order to find suited parameter sets for producing crack- and pore-free samples of the respective blends, a parameter study was carried out on each of the three blends. Optimal parameters were determined by varying the laser powder (250–400 W) as well as the scanning speed (400–800 mm/s).

**Table 1.** LPBF (laser powder bed fusion) parameters used to create the parameter study.

| Cube Size | Layer Thickness | Spot Size | Hatch Distance |
|---|---|---|---|
| $4 \times 4 \times 10$ mm | 0.05 mm | 0.05 mm | 0.08 mm |
| **Scanning strategy** | **Island size** | **Tilt angle** | **Gas** |
| Checkerboard | $1 \times 1$ mm | 12° | Argon |

The VED is calculated using Equation (1) where $P$, $v$, $h$, and $d$ are the laser power, the scan velocity, the hatch distance, and the layer thickness, respectively [37,38]:

$$\text{VED} = \frac{P}{v \times h \times d} \tag{1}$$

Although this value is often used in the literature, it should be known that it may not accurately reflect the effective energy transferred for melting. All values included in the formula are theoretical values, so it can be used as a design parameter to describe the LPBF process, but does not reflect the complex physics of the melt pool [39] or include material properties [38]. Likewise, this formula does not allow to draw conclusions on melt pool depth or shape, so very different melt pool morphology can be observed for the same VED [40]. Despite these drawbacks, VED is used here as a design parameter for comparison.

### 2.4. Phase Analysis

The quantitative measurements of phases in the microstructure were performed using XRD with Cr-K$\alpha$ radiation in a theta/theta diffractometer type Charon XL from XRD Eigenmann (Schnaittach-Hormersdorf, Germany). An angular range from 50° to 166° was measured with a step size of 0.05° with a location sensitive detector. The peaks {110} {200} and {211} of bcc ferrite/martensite and {111} {200} and {220} of the fcc austenite were recorded. The evaluation was carried out using the Rietveld method with a profile fitting of the entire diffraction diagram with the Topas 4.2 software from Bruker-AXS (Karlsruhe, Germany). Focus of the evaluation was lain on amount of phases e.g., retained austenite in the microstructure thus no instrumental broadening has been taken into account. Therefore no crystallite size and macrostrains could be derived from the measurements with accurate uncertainty. The analysed retained austenite fractions have an uncertainty of approx. $+/-$ 5 wt. %.

### 2.5. Microscopy

To investigate the particle morphology of the powders as well as the microstructures of the samples, a scanning electron microscope (SEM) (Vega II XLH, Tescan, Brünn, Czech Republic) was used. The SEM operated at an acceleration voltage of 15 kV, and at a working distance of 18–24 mm.

After the production, the specimens were eroded at half width, and hot embedding in backelit (Epomet, Buehler, Lake Bluff, IL, USA), mechanically grounded with SiC abrasive

paper (80–1200 mesh size), polished with a 3 μm polishing cloth (2TS1, Kulzer GmbH, Hanau, Germany) and a MetaDi Supreme for 10 min and finished with a one-minute fine polishing using a 0.02 μm polishing cloth (OP-Chem, Struers, Willich, Germany) and a Mastermet II (Buehler, Lake Bluff, IL, USA) suspension. Microstructure contrasting was conducted according to ASTM E407 using Nital (3% alc. HNO3). The porosity analysis of the microstructure was examined with an optical microscope (MZ16A, Leica, Wetzlar, Germany) using a 0.63× objective, a zoom factor of 1.903 and an eyepiece magnification 12.01:1. After analyzing the microstructural images with ImageJ (version 1.53a, NIH, Bethesda, MD, USA), these were evaluated these with regard to optical porosity analysis.

Element distributions were measured selectively with the EDS of the SEM (Vega II XLH, Tescan), but an electron probe micro analyzer (JXA-8200, JEOL, Akashima, Japan) was used for two-dimensional analyses. A measuring range of $700 \times 11$ steps with a step size of 10 μm was selected for large analyzing areas. For smaller sample sections, $100 \times 100$ steps with a step size of 1 μm or, for the line scan a line with $1 \times 100$ steps with a step size of 1 μm, was evaluated.

### 2.6. Hardness

For the mechanical test, a micro hardness testing was performed according to ISO 14,577 using a Fischerscope H100C (Helmut Fischer GmbH, Sindelfingen, Germany). A measuring field of $900 \times 900$ μm was selected and measured with a step size of 30 μm. A force of 50,000 mN was applied to all points in 10 s, held for 10 s and released within 10 s.

## 3. Results

The FeSiC system is a basic system for steels and cast irons and was chosen as a target alloy, due to several specific limitations and possibilities regarding phase and microstructure evolution in the LPBF process. With this alloy system, several effects like thermal history and liquid/solid solution dissolving can be investigated, which help to understand different process sequences in LPBF on the basis of chemical homogeneity and microstructure distribution. The system was limited to two important elements used in different steels and cast iron: Si and C.

In addition to the influence of the alloying elements on the microstructure evolution, the composition of the powder blends is also exerting great influence. Therefore, the blends are produced using different raw alloys (pure Fe, binary and ternary ferroalloys) so that their impact on the chemical homogeneity of the LPBF-manufactured samples can be investigated.

### 3.1. Powder Blending

Three different powder blends M1, M2, and M3 are prepared from the raw alloys (Fe, SiC, FeC, FeSi, and FeSiC) to achieve the same target alloy composition after LPBF processing (Table 2).

**Table 2.** Composition of the powder blends M1, M2, and M3 (wt. %).

|      | Fe    | SiC  | $Fe_{5.5}Si_{2.35}C$ | $Fe_7Si$ | $Fe_{4.3}C$ |
|------|-------|------|----------------------|----------|-------------|
| M1   | 95.00 | 5.00 | /                    | /        | /           |
| M2   | 36.39 | /    | 63.61                | /        | /           |
| M3   | 15.12 | /    | /                    | 50.00    | 34.88       |

Blend M1 consists of commercially available powders. Although, the used powders differ in their density (Fe: $7.874 \text{ g/cm}^3$, SiC: $3.21 \text{ g/cm}^3$). Furthermore, the SiC particles have an irregular, flat and sharp-edged shape owing to the applied milling process. In M2 and M3 the elements Si and the C are already dispersed inside one of the produced raw alloys. The main disadvantage of using such gas-atomized raw alloys lays within the fact that that these are currently not commercially available.

### 3.2. Powder Properties and Powder Morphology

The chemical compositions and the solidus and liquidus temperatures of the starting and target alloys are shown in Table 3. The solidus and liquidus temperatures of the binary and ternary ferrous-raw alloys are lower than that of the pure Fe and the SiC.

**Table 3.** Chemical composition and properties of the used powders.

|  | **Fe** | **SiC** | **Fe$_{5.5}$Si$_{2.35}$C** | **Fe$_7$Si** | **Fe$_{4.3}$C** | **Fe$_{3.5}$Si$_{1.5}$C \*** |
|---|---|---|---|---|---|---|
| Fe [wt. %] | 100 | — | >92.0 | balance | balance | 95.0 |
| C [wt. %] | — | 30 | 2.25 | — | 4.3 | 1.5 |
| Si [wt. %] | — | 70 | 5.20 | 6.87 | — | 3.5 |
| T$_{solidus}$ [°C] | 1538 | 2826 | 1167 | 1386 | 1153 | 1155 |
| T$_{liquidus}$ [°C] | 1538 | 3600 | 1207 | 1423 | 1158 | 1193 |

\* Target alloy composition.

As seen on the left side of Figure 1, the spherical Fe powder particles of M1 are surrounded by randomly distributed sharp-edged SiC particles. All powder particles in the blends M2 and M3 are highly spherical, with only few satellites attached to them.

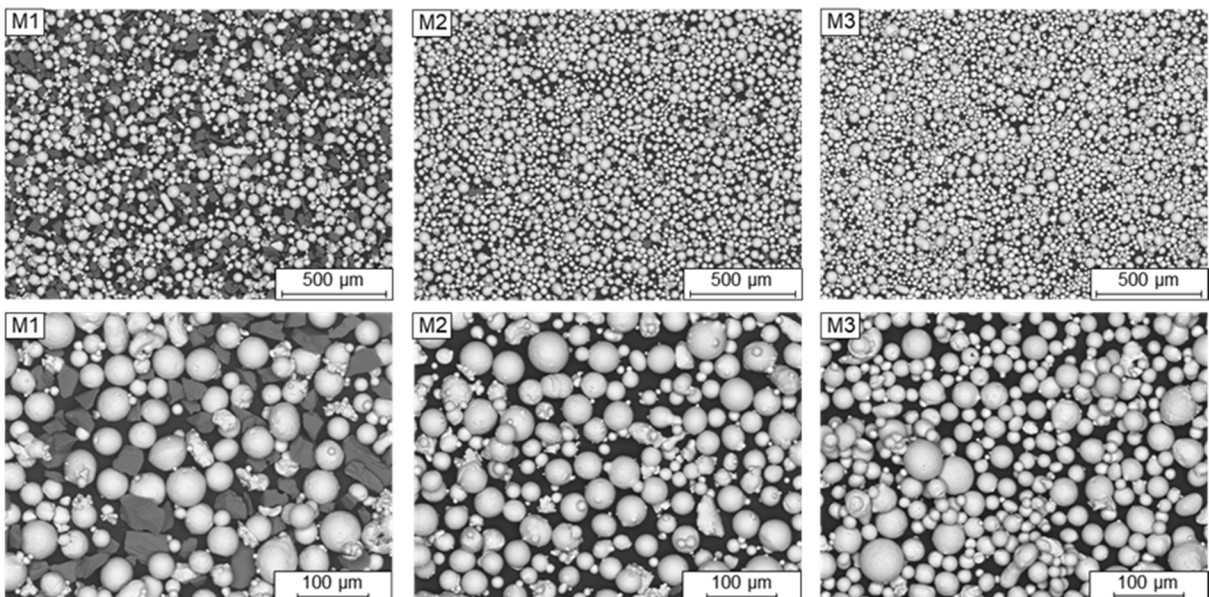

**Figure 1.** SEM (scanning electron microscope) powder morphology (**left**: M1, **middle**: M2, **right**: M3).

The particle size distribution of all raw alloys used to produce the blends can be seen in Figure 2. The pure Fe contains a small fine fraction, but a high proportion of particles >63 μm, which were not completely removed by the sieving process. SiC, Fe$_7$Si, and Fe$_{5.5}$Si$_{2.35}$C have a similar particle size distribution. Fe$_{4.3}$C has the narrowest distribution and thus the smallest fine and coarse fraction.

The resulting particle size distribution of the powder blends are shown in Figure 3. M1 and M3 show a shift to the right compared to M2 and thus tend to have a higher number of larger particles. The table on the right shows that M1 and M3 contain at least 10% particles >63 μm. This can be attributed to the high proportion of larger particles in the Fe-powder (Figure 2).

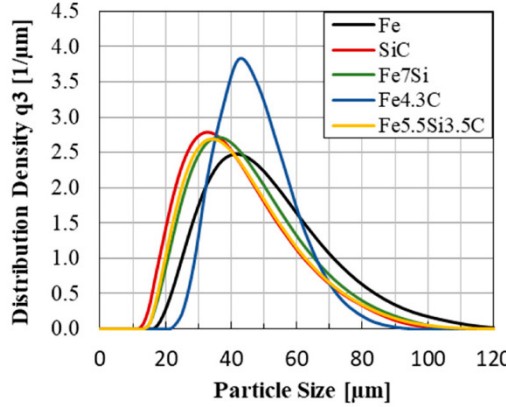

| Particle size distribution [μm] | d (0.1) | d (0.5) | d (0.9) |
|---|---|---|---|
| Fe | 31.4 | 50.0 | 78.2 |
| SiC | 22.0 | 37.2 | 61.5 |
| Fe5.5Si3.5C | 23.7 | 39.4 | 68.6 |
| Fe7Si | 24.4 | 40.0 | 64.4 |
| Fe4.3C | 31.6 | 43.4 | 59.3 |

**Figure 2.** Particle size distribution of all used raw alloys (**left**: diagram, **right**: numbers).

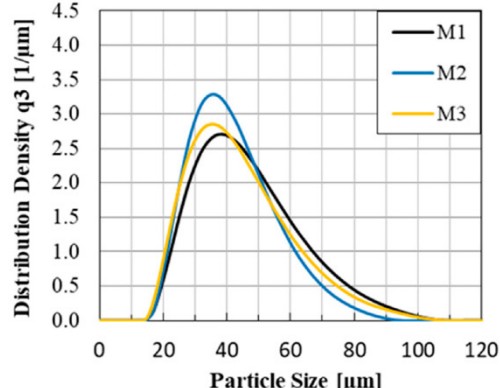

| | M1 | M2 | M3 |
|---|---|---|---|
| **Particle size distribution [μm]** | | | |
| d (0.1) | 25.8 | 24.7 | 24.2 |
| d (0.5) | 41.6 | 37.8 | 39.1 |
| d (0.9) | 65.7 | 57.2 | 62.8 |
| **Flowability** | | | |
| Hall flow [s/50 g] | 22.8 | 17.0 | 28.9 |
| Hausner Ratio | 1.12 | 1.16 | 1.23 |

**Figure 3.** Powder blends characteristics of M1 (black), M2 (blue) and M3 (yellow) (**left**: diagram, **right**: numbers).

The Hall flow tests and Hausner ratio indicate good flow properties according to Carr et al. [35] for M1 and M2. Considering the Hausner ratio, M1 showed the best flow properties and in the Hall flow its value stood between the other two blends. In contrast, M3 indicates poorer flowability but is still rated fair according to Carr et al. [35].

### 3.3. Characterization of Densified Specimens

Due to the different chemical composition and melting temperatures of the particles in the used powder blends, a parameter study was carried out to determine the optimum LPBF parameters for each blend. The parameter sets of 400 W and 700 mm/s (M1) and 250 W and 700 mm/s (M2 and M3) resulted in the highest relative density. These samples were used for the subsequent investigations.

Due to the high Si-content of the target alloy, which leads to lower melt surface tension and lower kinetic viscosity [41,42], there was increased spatter formation during the LPBF process of all powder blends. This spattering was highest in blend M1, but was evident in all three blends. This limits the general usability of the target alloy and is therefore independent of the selected raw alloys and blending strategies.

In the following, the additively generated samples from M1 are named M1*, from M2 to M2* and M3 to M3* to make the difference between the powders and the densified specimens explicit.

### 3.4. Porosity

The LPBF-densified samples were cut at half of the cuboid, as shown on the left in Figure 4, and then analyzed by means of light optical analysis. All three specimens possess a relative density above 99%. This shows that the LPBF processing of such powder blends is technically feasible.

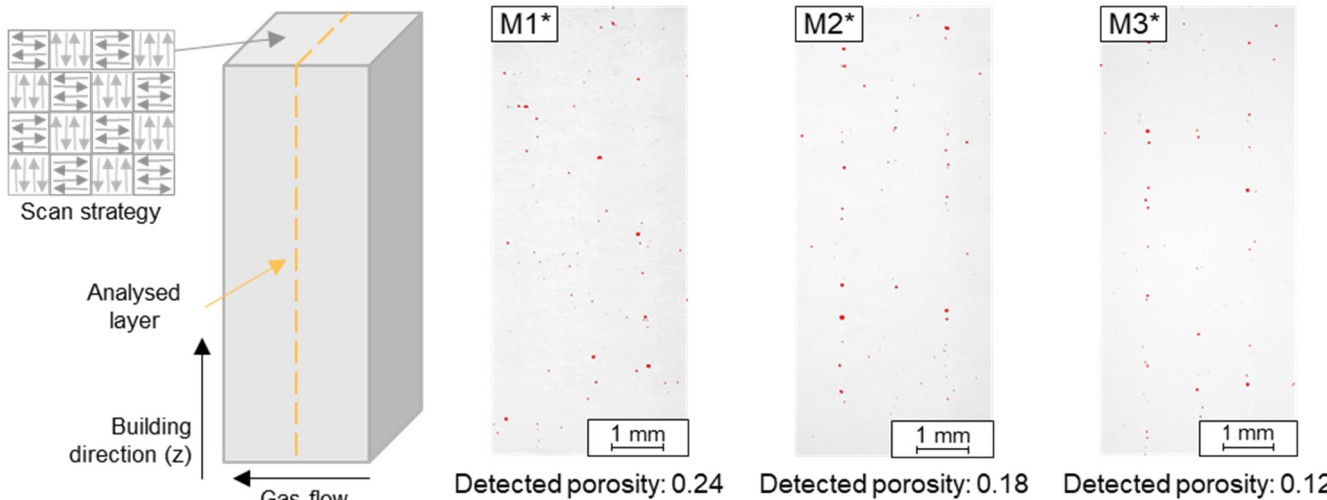

**Figure 4.** Microscopical image analysis of the optical porosity of M1* (**left**), M2* (**middle**) and M3* (**right**) (the as porous detected areas are highlighted in red).

The cross-sections of the densest specimens LPBF-produced from the respective powder blends can be seen in Figure 4 on the right side. The porous areas are highlighted in red. A porosity of 0.24% was achieved with M1, 0.18% with M2 and 0.12% with M3. Thus, when comparing the blends, the densest structure could be produced with M3. All three samples show a linear pore pattern.

### 3.5. Microstructure

The alloy consists mainly of Fe as matrix element and the substitutional alloying element Si (3.5 wt. %) and 1.5 wt. % C as interstitial. The effective amount of 1.5 wt. % C in the alloy has a strong effect on the martensite start temperature ($M_S$). From the empirical formula of Barbier [43], see Equation (2), an $M_s$ of 59 °C can be calculated.

$$\begin{aligned} M_s ={}& 545 - 601.2(1 - \exp(-0.868\,\text{C\%})) - 34.4\,\text{Mn\%} - 13.7\,\text{Si\%} - 9.2\,\text{Cr\%} - 17.3\,\text{Ni\%} - 15.4\,\text{Mo\%} \\ & -2.44\,\text{Ti\%} - 361\,\text{Nb\%} - 1.4\,\text{Al\%} - 16.3\,\text{Cu\%} - 3448\,\text{B\%} + 10.8\,\text{V\%} + 4.7\,\text{Co\%} \end{aligned} \tag{2}$$

Figure 5 shows sections of the microstructures of all three samples in the overview images on the left. It can be seen that different microstructures were produced and the individual microstructures are inhomogeneous. In M1* on the left, one can see that there are many different etching areas, so that the microstructure seems to be locally rather inhomogeneous. The black areas seem to be unmolten particles, which are clearly visible at higher magnification. A more detailed examination of the structure reveals that the lighter areas have a needle-like structure which is associated with a martensitic microstructure which must feature according to $M_s$-calculations retained austenite. In comparison, the dark grey areas (second picture) can be identified as more refined structures being associated with a mixture of bainite (according to the morphology of microstructure features), retained austenite, and only little martensite.

In the overview of M2*, the microstructure appears more homogeneous, but also shows locally differing areas. Though in fewer number, unmolten particles can also be seen. A closer look at the microstructure reveals bright and long-needled areas predominantly with a martensitic microstructure in the vicinity of unmolten Fe particles. Additionally, the refined-looking darker areas are composed mostly of carbide free bainite and retained austenite microstructure constituents. In the vicinity of unmolten particles, a seam of perlite in the particle could be revealed (see Figure 5 upper right microstructure picture of M2*).

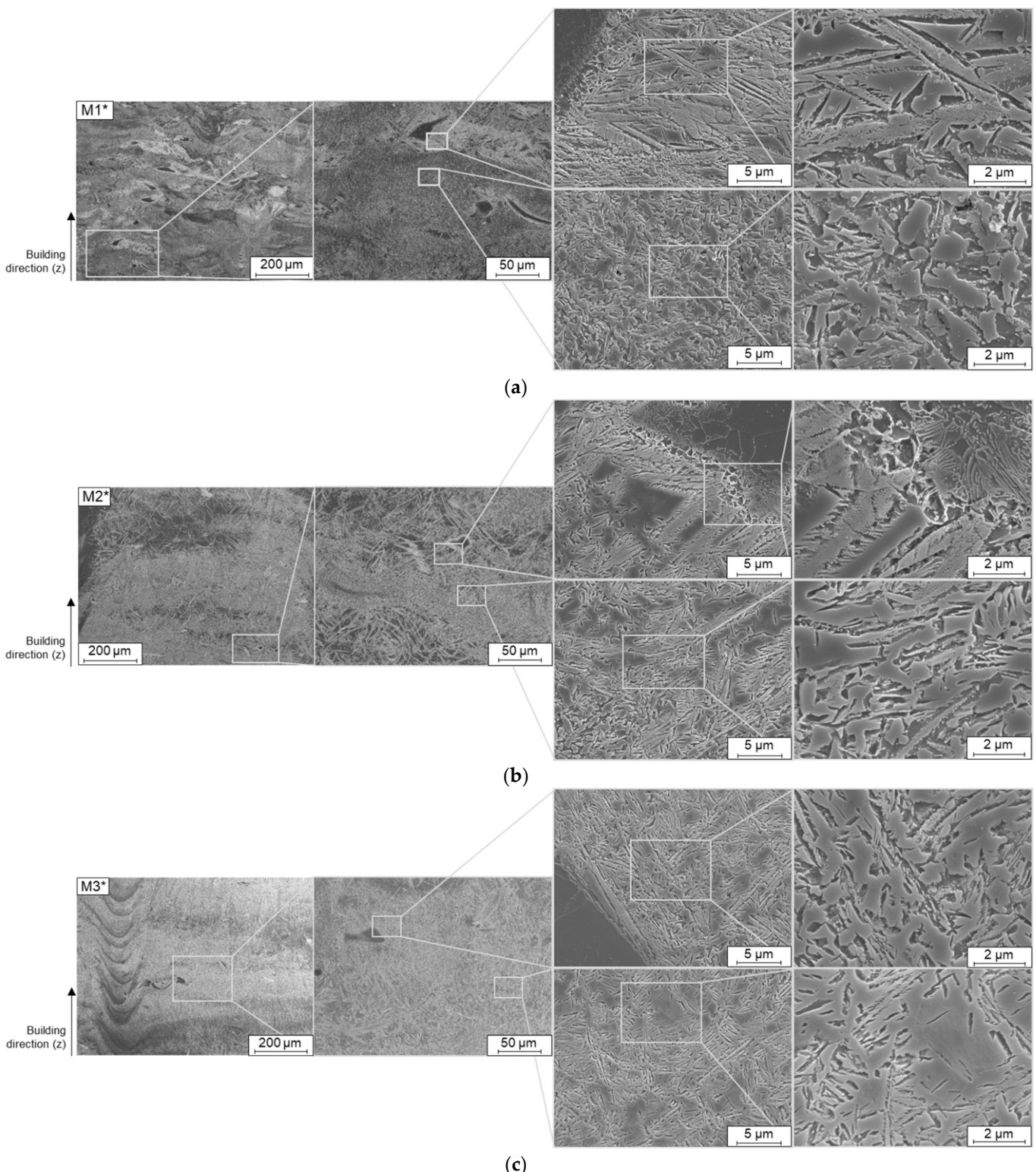

**Figure 5.** SEM microstructure of (**a**) M1*, (**b**) M2* and (**c**) M3*.

M3*, on the other hand, appears finer and more uniform than the other two microstructures, but also shows a few unmolten particles. The detailed images of the microstructures confirm this impression and show a fine needle-like, mostly carbide-free bainitic microstructure with retained austenite present.

### 3.6. Chemical Composition and Homogeneity

When examining the individual microstructures by means of SEM, unmolten particles were noticed in all blends (Figure 6). With a closer examination by EDS these particles could be identified as not fully molten, pure Fe particles. In Table 4 it can be seen that the Si and C contents at the locations marked with 1 drop down to 0 wt. % and the Fe content rises to almost 97 wt. % in all three used blends. Around the non-melted particles at the marked points 2–4, the Fe and Si contents fluctuate evenly.

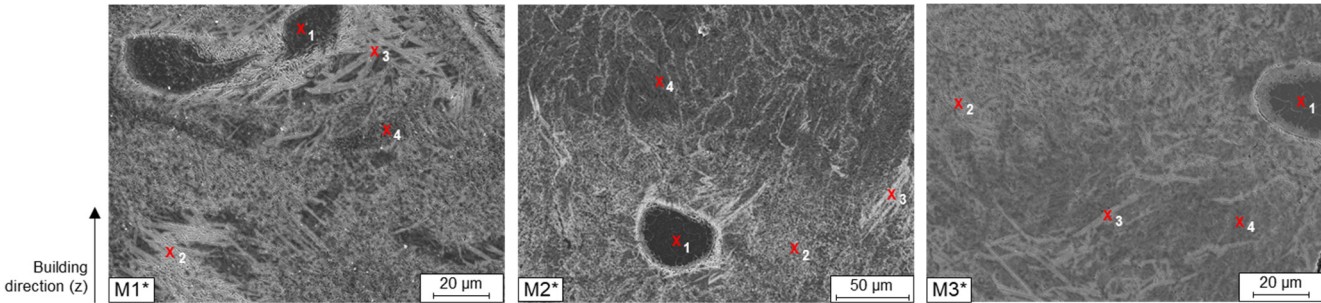

**Figure 6.** EDS analyzation of microstructure (1: unmolten particle, 2: light area, 3: white needle-like area, 4: dark area).

**Table 4.** EDS measurement corresponding to the microstructures shown in Figure 6.

| | M1* | | | | M2* | | | | M3* | | | |
|---|---|---|---|---|---|---|---|---|---|---|---|---|
| Point | 1 | 2 | 3 | 4 | 1 | 2 | 3 | 4 | 1 | 2 | 3 | 4 |
| Si [wt. %] | 0.00 | 1.43 | 1.99 | 3.26 | 0.00 | 2.30 | 2.10 | 2.29 | 0.00 | 2.40 | 2.25 | 2.43 |
| Fe [wt. %] | 96.65 | 94.00 | 92.37 | 93.06 | 96.80 | 94.00 | 92.99 | 94.34 | 96.64 | 93.02 | 90.85 | 93.53 |

The particles seen in Figure 6 are partially melted during the short residence time of the melt pool during the LPBF process. This can be seen in the stream-lined drop shape of the remaining Fe particles in M1*. The diffusion zones around the partly molten particles can be identified as bright areas in the SE contrast. Near the unmolten Fe-particles a predominantly bainitic microstructure is formed owing to the locally decreased C and Si content (Table 4). The re-melting of an already solidified layer will happen twice at a maximum. Therefore, the Fe-particle act like Si and C content drops for the surrounding matrix, depleting predominantly C in their surroundings. Consequently, a changed microstructure morphology can be seen in the transition zones. Additionally, one can observe an internal seam in the unmolten Fe-particles due to C diffusing into the Fe particle during the LPBF process, increasing the local C content. Because of the carbon lacking in the Fe-particle a mainly ferritic microstructure can be observed (Figure 7) in the particle and a seam of bainite and perlite is formed due to the carbon diffusing into the particle from the melt and solidified surrounding material. This area then is suffering from a carbon depletion and is showing a different transformation behavior either.

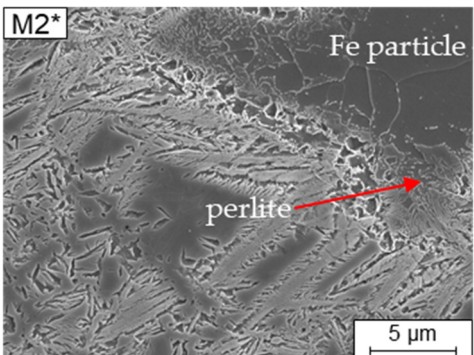

**Figure 7.** Incompletely molten Fe-particle in alloy M2* (SEM-SE).

During cooling a lower bainite microstructure with carbides is achieved. Because of the intrinsic heat treatment [44] by adding new layers being molten by the laser, the temperature interval of the solidified material is kept for some time in the temperature field of the bainitic transformation. Thus, bainite not only is formed during continuous cooling but also isothermally during the subsequent time when the temperature is kept within the temperature field of bainitic transformation because of the intrinsic heat treatment occurring during layered manufacturing.

The microstructure over all is not perfectly homogeneous. Because of the melt pool movement, it could be seen, that with changing direction of laser movement at each exposed layer a predominant microstructure alignment has happened due to directed solidification. Besides, it seems to make a distinct difference if already solidified material is remolten and solidified and then intrinsically heat treated afterwards. Therefore, a mixed microstructure is derived consisting of martensite, austenite and bainite in the matrix inheriting some directed orientation from laser passage during exposure. Additionally, microstructure is impacted by chemical inhomogeneity that mainly appear in the vicinity of poorly or unmolten Fe-particles.

To determine the local chemical element distribution in an area surrounding the unmelted Fe particles, micro probe mappings (wavelength dispersive x-ray spectroscopy) were carried out (Figure 8). The Fe particles are the only particles that were not dissolved during the LPBF densification. These can be found in all three samples. Figure 8 shows a line scan (plotted in orange) of an example particle from M3*, where it is visible that that the Fe value on such a particle increases, but the Si and C values decrease to 0 wt. %. This eliminates the possibility that it is anything other than a pure Fe particle. The line scan also shows that around the particle the element distribution of all three elements fluctuate but are present everywhere. Compared to the other elements, the C value varies strongly between 0.5 and 4 wt. % in the environment of an unmelted particle.

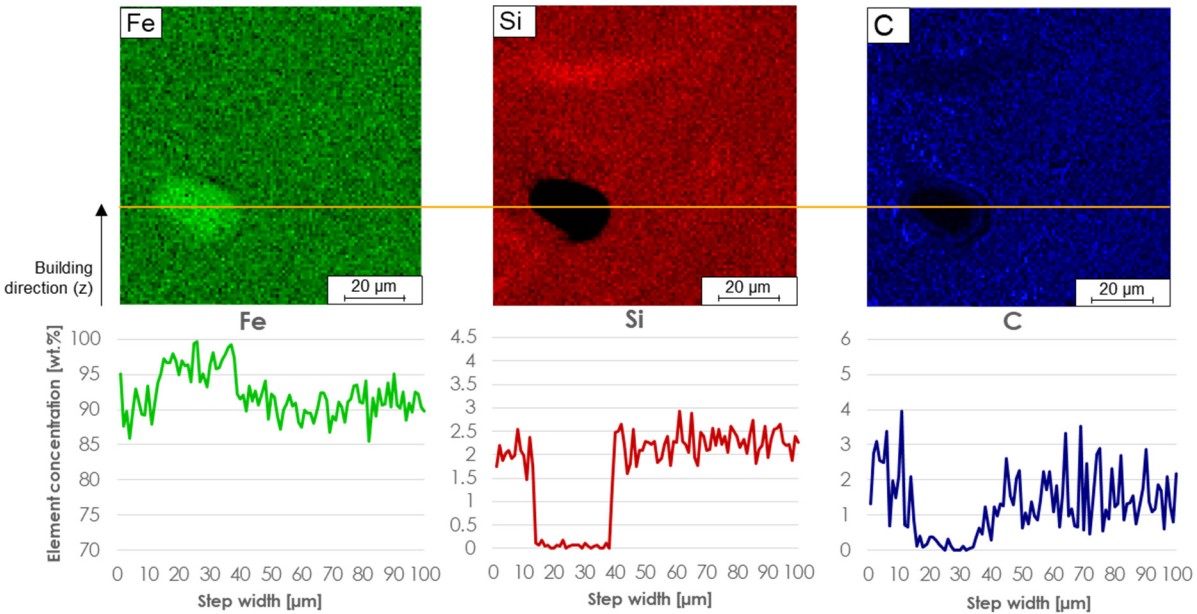

**Figure 8.** Microprobe results of an unmolten particle in M3* (measured line: orange).

This also means that the energy introduced was not sufficient for all Fe particles to be completely melted. Most particles were detected in M1*, although a VED of 143 W/mm$^3$ was introduced there. In comparison, 89 W/mm$^3$ was used for M2* and M3*. Due to the higher number of unmelted particles in M1* and the amount of 95% of the Fe raw alloys in the M1 blend, this indicates that too many larger pure Fe particles >63 μm were present in the blend despite the higher energy applied. Likewise, the size of the particles seen in Figure 8 indicates that the Fe particles of the feedstocks must have been too large before

the LPBF process, otherwise the particles would have dissolved completely. To avoid unmelted Fe particles, a higher energy input can be used or Fe particles >63 μm must be avoided. Above the unmolten Fe particle in the Si-distribution mapping in Figure 8 an additional inhomogeneity can be seen. There is an area with an arc like distribution of Si which is increased above average value in the image. Deriving from the above mentioned inhomogeneity it is recommended to use smaller particles instead of increased energy input, because higher energy input can lead to other defects, e.g., balling effects or so-called keyhole pores [9,45–47].

As the microstructures show local chemical inhomogeneities, the microprobe investigation was used to obtain a value to describe this issue. Thus, these investigations are carried out to develop a methodology and a value to determine the blends and produced alloys in terms of their quality. The deviation of the element distribution is used as a measure to draw conclusions on the overall homogeneity. The element with the highest sensitivity of the measurements with respect to the element distribution is used as the evaluation criterion. In this case, the distribution of Si is used as a measure of homogeneity in the microstructure, as it shows the highest sensitivity because of the low diffusion coefficient and therefore only limited mobility by the means of diffusion after solidification (Figure 9) [48].

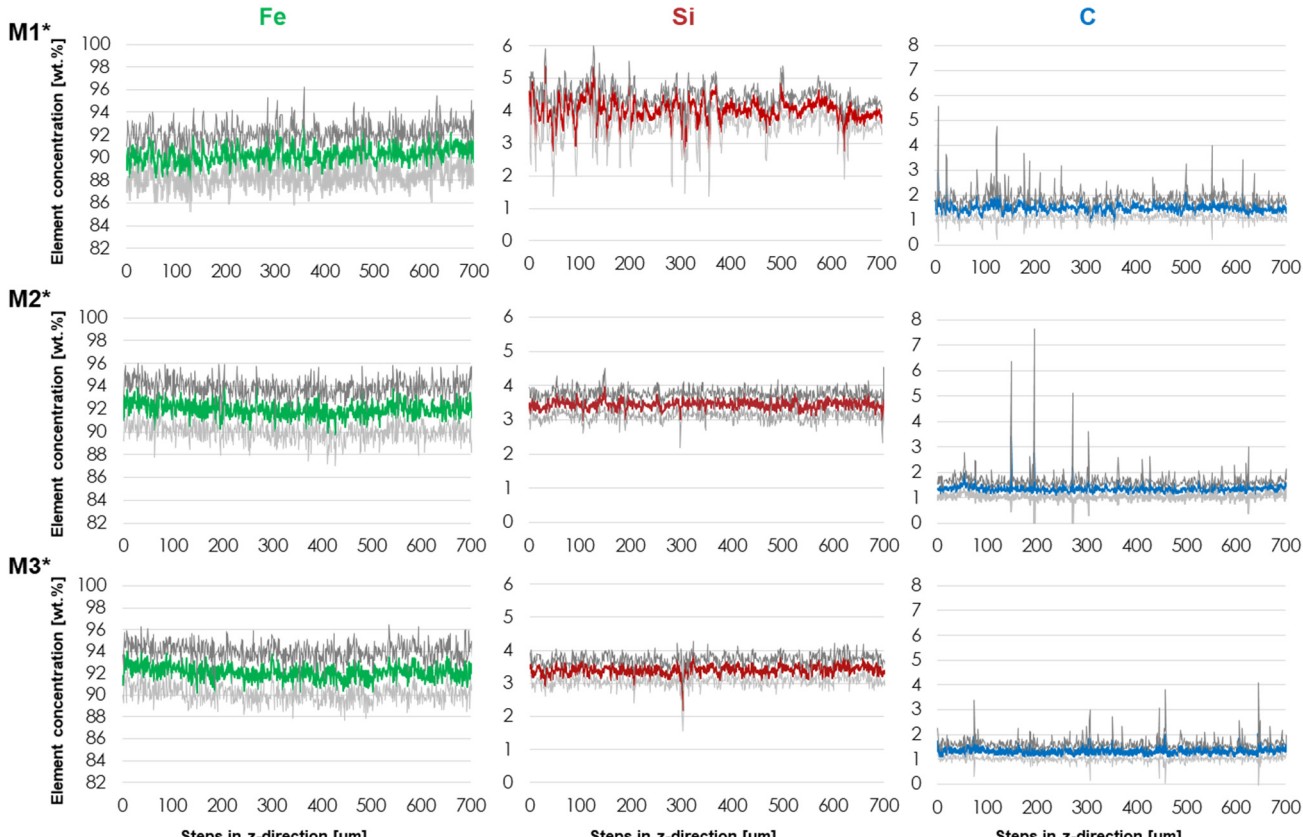

**Figure 9.** Microprobe results for M1*, M2* and M3* (green: mean Fe, red: mean Si, blue: mean C, dark grey: mean plus standard deviation, light grey: mean minus standard deviation).

For the evaluation, a range of 11 measurement points was selected for all samples in which no pores or defects were present, so that possible deviations can be excluded. The green lines in Figure 9 show the mean value for Fe, the red lines refer to the mean value of Si and the blue lines indicate that of C. The mean value of the 11 measuring points is always shown over one measuring step in the z-direction so that the local fluctuations in z-direction can be displayed. The dark grey line shows the mean plus the standard deviation;

the light grey represents the mean minus the standard deviation of the 11 measured values per z-measurement step.

As can be seen in Figure 9, the microprobe measurements of the individual elements show differences regarding the LPBF samples examined from the different blends. The pure Fe content varies similarly for all of them, yet around a mean value of about 90 wt. % for M1* and 92 wt. % for M2* and M3*. The Si content varies more significantly for M1* than the other two blends. For M2* and M3*, the values vary more evenly around a value of 3.5 wt. %, with M3* showing a peak towards 2 wt. %. For C, all blends vary on average around a value of 1.5 wt. %, but the variations differ for the individual blends that are used. For M1*, the fluctuations are highest over the entire sample length; but for M2*, individual measuring points are clearly above the remaining deviation of the entire sample. In M3* there are also individual peaks, but they are smaller in comparison and thus shows the smallest variation.

In order to evaluate which powder blend is most suited to be densified by LPBF from these diagrams, the value of the standard deviation over the complete measuring range is used as the basis for evaluation. The smaller the standard deviation and thus the local variations, the more homogeneous the element distribution of the overall structure. Table 5 shows the mean values and standard deviations of the entire sample cross-section of the measurements shown in Figure 9. It is proven that the mean value of M1* is lowest for pure Fe, but highest for Si and C. M2* and M3*, contrastively, are similar with regard to the mean value of all elements. The highest deviations in the standard deviation are also found for M1*. For M2* and M3*, the standard deviation of Fe and Si are almost identical. However, the standard deviation of the C content is higher for M2* than for M3*. Since the C content can cause the formation of martensite, graphite or carbides, the evaluation here is based on the most sensitive element, which is the Si content. The variation of the standard deviation around the mean is used as a criterion. Consequently, there is a fluctuation of 15% around the mean value for M1* and 11% for M2* and M3*. Also, the quality in terms of homogeneity of M2* and M3* are similar, but M1* can be described as more inhomogeneous.

**Table 5.** Mean values and standard deviations [in wt. %] of the microprobe analysis.

|  | **M1*** | **M2*** | **M3*** |
|---|---|---|---|
| **Mean value [wt. %]** |  |  |  |
| Fe | 90.2 | 92.0 | 92.1 |
| Si | 4.0 | 3.4 | 3.4 |
| C | 1.5 | 1.4 | 1.3 |
| **Standard deviation [wt. %]** |  |  |  |
| Fe | 2.2 | 2.1 | 2.1 |
| Si | 0.6 | 0.4 | 0.4 |
| C | 0.5 | 0.4 | 0.4 |

### 3.7. Phase Analysis

The measurement of the effective amounts of phases was performed using XRD. Because Rietveld refinement was used information about the lattice parameters, the associated errors and the quality of the refinement are given in Table 6. The goodness of fit (GOF) and $R_{wp}$ show that the fit is good.

The results of the XRD (Figure 10) reveal the powder blends' impact on the final retained austenite content although all powder blends have a rather similar chemical composition. The target alloy $M_S$ temperature (around 60 °C) promote the existence of retained austenite. M1* shows the least retained austenite content with $20 \pm 5$ wt. %. M2* contains about 34 wt. % and M3* 37 wt. % retained austenite. The differences between the samples are most likely bound to the inhomogeneity of the chemical element distribution in the manufactured specimens. Si and C will impact the bainitic transformation in its kinetics.

The retained austenite content, accordingly, is highest for the sample of M3*, which contains the highest chemical homogeneity and the most homogenous microstructure distribution.

**Table 6.** Crystallographic parameter derived from Rietveld refinement.

|  | Lattice Parameter Austenite [nm] | Error [nm] | Lattice Parameter a Martensite [nm] | Error [nm] | Lattice Parameter c Martensite [nm] | Error [nm] | Goodness of Fit (GOF) | $R_{wp}$ |
|---|---|---|---|---|---|---|---|---|
| **M1*** | 0.36191680 | 0.0001190 | 0.2857898 | 0.00007540 | 0.2866975 | 0.0000629 | 1.44 | 2.05 |
| **M2*** | 0.36176400 | 0.0001669 | 0.2857726 | 0.00011088 | 0.2867903 | 0.0000876 | 1.36 | 1.85 |
| **M3*** | 0.36188687 | 0.0002036 | 0.2857206 | 0.00013450 | 0.2868139 | 0.0001090 | 1.40 | 1.86 |

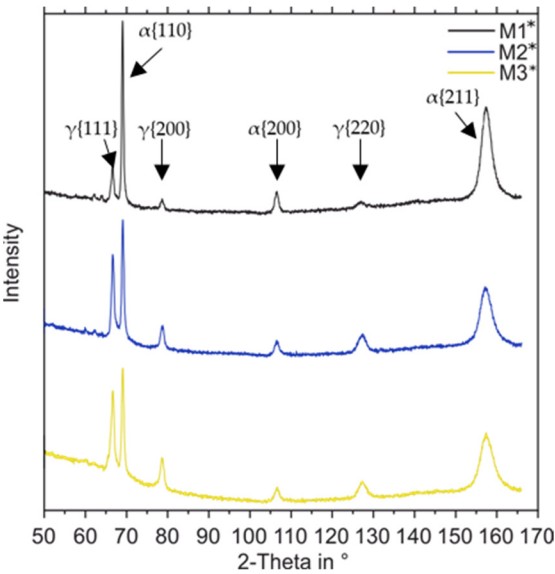

**Figure 10.** Phase analysis (M1: black, M2: blue, M3: yellow).

### 3.8. Hardness

In order to be able draw conclusions from the previous investigations on the mechanical properties which are related to the microstructures and to evaluate the influence of the inhomogeneous element distribution, micro hardness mappings were conducted on the samples. As shown in Figure 11, the micro hardness maps are measured in an area of 900 × 900 μm with a step size of 30 μm. The corresponding measured values can be found in Table 7.

M1* stands out with its highest mean value of about 741 HV0.005, but shows only isolated harder local spots, so that a standard deviation of around 61 HV0.005 is present. It is visible that although the elemental homogeneity of M2 and M3 were similar, the hardness measurement results are different. The mean value is again similar for both (703.58 and 707.86 HV0.005), but the standard deviation for M2 of 84.73 HV0.005 is higher than for M3 with 60 HV0.005. M2 stands out due to the significant increases in local hardness of up to 1163 HV0.005. M3, on the other hand, appears most uniform and has isolated hard areas only in the region of the right edge.

**Table 7.** Results of the micro hardness tests.

| [HV0.005] | Mean Value | Standard Deviation | Maximum Value |
|---|---|---|---|
| **M1*** | 740.8 | 60.9 | 1035.1 |
| **M2*** | 703.6 | 84.7 | 1163.3 |
| **M3*** | 707.9 | 60.2 | 964.6 |

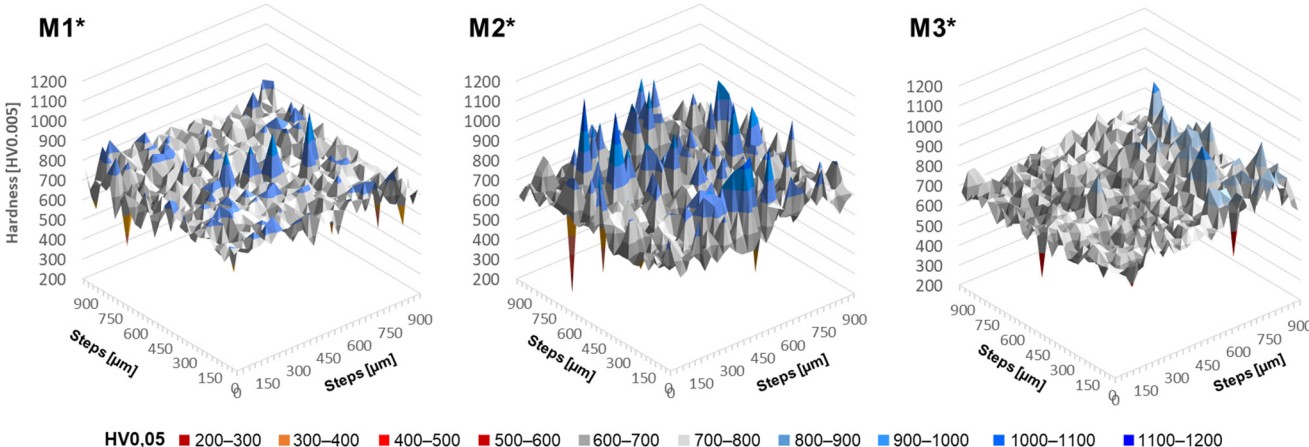

**Figure 11.** Micro hardness surfaces of M1*, M2* and M3*.

In order to assign the local differences to the microstructures, the measured areas were examined in more detail by SEM, here using M3* as an example. The region possessing the average hardness as well as that with the highest hardness (directly on the edge of M3* in Figure 11) were investigated. In the area possessing the average hardness (Figure 12a), a uniform, fine, acicular structure can be found, which indicates carbide-free bainite and austenite. On the other hand, within the harder area, as seen in Figure 12b as a darker area, martensite and retained austenite could be identified.

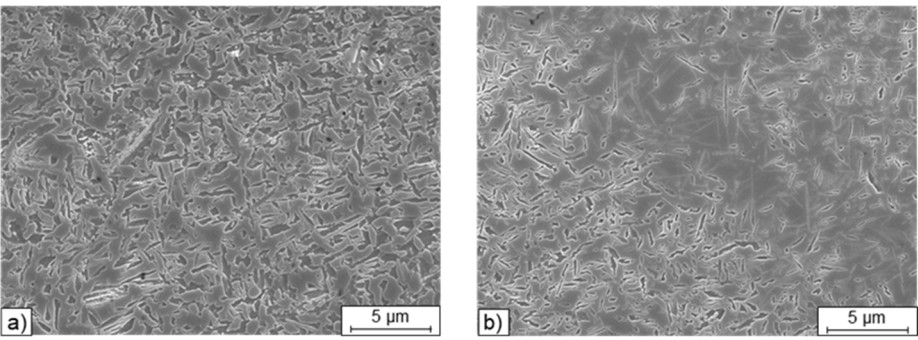

**Figure 12.** SEM of the microstructure of M3* from Figure 11 around a mean (**a**) and a harder region (**b**).

## 4. Discussion

### 4.1. Powder Blending

The Table 3 of the powder blends show the blending ratios used. In the LPBF process, due to the high liquidus temperature of SiC (about 3600 °C ) and the necessity of melting and dissolving the elements into the liquid Fe by a mostly parallel solid to liquid dissolution process at the phase boundaries, inhomogeneous element distribution or incompletely melted SiC particles may occur. Bigger particles may not dissolve during the short residence time of the melt pool during LPBF processing. The binary and ternary raw alloys in M2 and M3 are possibly more suitable for the LPBF process. Based on the different liquidus and solidus temperatures of the elements and raw alloys, it can be assumed that the blends will melt differently in the melt pool during LPBF. Due to the lower liquidus temperatures, it can be assumed that a more homogeneous microstructure can be achieved with M2 and M3. Because the melting and mixing process in the melt pool is complex and the melt pool movement bears different effects, such as diffusion, Marangoni, and various mechanical movement effects, these are only a few possible influencing factors on how differently the blends might behave in the LPBF process.

### 4.2. Powder Properties and Powder Morphology

The morphology of powders has an impact on the characteristic powder properties like flowability [49]. For a dense and defect-free sample, it needs to be possible to apply a dense and uniform powder layer with a doctor blade. As seen in Figure 1, there is no restrictions in flowability and processability expected due to the spherical particle morphology. This was also proven by the flowability tests (Table 3) and the obtained samples. A negative influence on the flowability of the broken SiC particles in M1 could not be detected, but rather the opposite. The milled SiC powder seems to improve the flow behavior in the added amount, based on the shape and surface of the particles. The reason for the lower flowability of M3 must be clarified by subsequent investigations. The FeSi showed poorer flow behavior, despite the same morphology and particle size distribution in regard to the other raw alloys. It is currently assumed that the specific surface characteristics of the FeSi particles could be a reason for the changed particle interaction and thus the flowability. For this purpose, high-resolution scanning electron microscopy images will be used in future investigations.

### 4.3. Characterization of Densified Specimens and Porosity

LPBF processing of all investigated powder blends was possible at very low porosity but by using different parameter sets. According to Equation (1) with the optimal used parameter set, M1 required a VED of 143 W/mm$^3$ for the production of a dense sample by LPBF, while M2 and M3 required 89 W/mm$^3$. This shows, among the reasons for the different liquidus temperatures of the raw alloys, that different energy inputs are required to melt the powder blends uniformly to produce a high relative density in the generated microstructure.

Figure 4 showed, that all three samples show a linear pore pattern, which can be explained by the turning points in the checkerboard scanning strategy. These pores are so called keyhole pores, which result from the inclusion of vaporized metal during the solidification of the melt, caused by an excessively high local laser energy input [47,50]. At the point where the laser reaches the end of a linear scan path or the turning point of the line scan, the laser does not reduce the energy and additional energy is added, resulting in overheating and vaporization of some of the metal in the melt pool. When the laser moves away from this point, the vapor depression collapses and a gas pore is introduced in the metal [51]. This pore formation can be avoided by using a different laser scan strategy (e.g., line scan) or by adjusting the laser power to keep the energy density approximately constant.

### 4.4. Microstructure

From the microstructure perspective the high amount of Si hinders the formation of Fe carbide in the material and promotes the ability to achieve solidification in the stable Fe-graphite system. Because of the rapid solidification rate associated with LPBF, solidification could be expected in the meta-stable system of Fe-cementite. The specific LPBF parameters, such as the specific VED used, can influence the overall temperature distribution and thus also the cooling rates, so that a change from the stable system into the metastable system is possible. In addition, the specific effect of Si, which suppresses the carbide precipitation, makes the Si distribution in the alloy traceable from the microstructure formation. Areas of low Si tend to transform non-martensitic due to low hardenability or into carbide bearing bainite because of the lack of suppression of carbide formation at low Si contents.

By the addition of 3.5 wt. % of Si a carbide-free bainite is formed which will mostly be hampered in its growth kinetics by C accumulating around the mobile interface. As a result, a full bainitic transformation is rather unlikely to happen, leaving stabilized retained austenite in the microstructure (Figure 5).

Besides Si the second alloying element is C, which can be redistributed in the melt and subsequently during the cooling phase by diffusion in the solid. C is a major alloying element for steels and cast irons, and is used for the formation of carbon martensite, carbides

or graphite (grey cast iron). The 1.5 wt. % C in the alloy has a strong effect on the martensite start temperature ($M_S$) so an $M_s$ of 59 °C (Equation (2)) can be calculated. There is a lack of data on the impact of high Si content on the $M_s$-temperature, accordingly, some deviation of the calculated $M_s$-temperature should be taken into account. C, as a major hardenability impacting element in the alloy, will impact local phase distribution and can be decisive if there is a bainitic or martensitic transformation as well as on the final retained austenite content. The solidification melt, which becomes austenitic at temperatures >911 °C , and the resulting microstructure are influenced by unmelted particles. The fast-diffusing element C can segregate at the peripheral zones of such particles owing to the local chemical potential. Due to the lack of Si there, a bainitic or perlitic microstructure can form on such particles.

The microstructure distribution in the samples is rather inhomogeneous considering the distribution of martensite, bainite and retained austenite. The local inhomogeneity is driven by three main effects: first the intrinsic heat treatment and re-melting of solidified layers is impacting the local morphology of martensite and austenite. As a second effect the retained austenite formed during last thermal hysteresis reaching temperatures above Ac1 is impacted by the intrinsic heat treatment because of the layered buildup of samples. Because of the re-heating of prior solidified layers austenite can transform into bainite during the re-heating phases as a result of thermally induced diffusion governed transformations. This effect is additionally super posed by the effect of heat dissipation in the base platform. Resulting from this effect there is a microstructure gradient all over the sample with increasing distance from the plate. The third effect adding some inhomogeneity to the derived microstructure is introduced by local chemistry. Unmolten particles and inhomogeneous distribution of elements coming from big pre-alloy particles or particles with increased melting enthalpy which could not be homogenized in the melt pool impact transformation kinetics during the austenite to bainite transformation due to intrinsic re-heating. Therefore, local austenite content stabilized by the primary alloy derived from solidification and effects coming from C partitioning are impacting the homogeneity of the samples. According to the results derived from the experiments the most detrimental effect on homogeneity appears to be the particle size of pure Fe used for dilution of all other raw alloys.

*4.5. Chemical Composition and Homogeneity*

The micro probe analyses (Figure 8) showed an individual high C contents of M2*, which could indicate carbides that may have formed during the LPBF process. The variation in the C content influences the local microstructure formation of bainite, perlite and carbides significantly. But it should be mentioned that a chemical homogenization and thus a transformation of the obtained microstructure can be achieved by subsequent thermal post-processing like annealing or HIP.

As seen in the picture from M2* (Figure 8), the Si and C content varied. Above the orange line there is a brighter arc like distribution of Si, and in the same area the C content is lower (the area is darker). This specific inhomogeneity is most likely achieved from a big FeSi pre-alloy particle. Because of the arc like distribution it can be derived that the local melt pool must have been enriched with Si due to a big but fully molten FeSi particle. Adjacent remelting of the next layer then has not fully inclined the Si-enriched area and therefor has left some of the increased Si-content in the matrix. Besides, particle size can be considered decisive for resulting homogeneity. Because of fast melt pool solidification and melt pool size a particle of FeSi of 60 μm could locally increase Si-content. If the re-melting during subsequent exposure is not inclining the full melt pool of adjacent solidification remainders of local inhomogeneity will remain as arc like artefacts.

Microstructural inhomogeneities can be seen in the micrographs and the compositional analysis. These local inhomogeneities that cannot be considered always as defects for all applications. The incomplete dissolution of the admixed particles may provide an opportunity to produce multi material structures like wear-resistant metal matrix composites (MMC) [52,53]. These properties can become a key factor for certain applications

and thus improve local functional properties. If actually dissolvable particles, which do not dissolve completely in the LPBF process, are used selectively, this can be used to advantage when thinking of soluble hard materials such as carbides, for example. In this way, hard alloys/MMC with specifically adjusted (large) hard phases can be achieved. For the Fe particles, a detrimental effect on the strength and fatigue of such a sample has to be considered.

### 4.6. Phase Analysis

The target alloy exhibits a rather low $M_S$ temperature, promoting the existence of retained austenite. Because of $M_s$ being around 60 °C it is rather likely to achieve a temperature where martensite is formed only in the beginning of the process. Additionally, the amount of martensite being formed must be rather low as room temperature is just 40 °C below $M_s$. Therefore, less than 50% martensite could be expected from solidification to cooling to room temperature. Furthermore, it must be taken into account that during layered manufacturing the insulating effect of the powder and accumulated heat will increase the temperature above $M_s$ during building. Then, this process changes the transformation from a martensitic into an isothermal bainitic transformation.

In addition, the effect of C partitioning from martensite to austenite after first solidification and subsequent intrinsic heat treatment and the C partitioning to austenite during bainitic transformation at areas of high Si contents can locally increase the C content in the phase austenite. Therefore, most stable austenite is expected for the sample with the best chemical homogeneity and highest amount of evenly distributed bainite [54,55]

The differences in the content of the retained austenite between the samples (M1*: $20 \pm 5$ wt. %, M2*: 34 wt. % and M3* 37 wt. %) are most likely bound to the inhomogeneity of the chemical element distribution in the manufactured specimens. Si and C will impact the bainitic transformation in its kinetics. The retained austenite content, accordingly, is highest for the sample of M3*, which contains the highest chemical homogeneity and the most homogenous microstructure distribution.

### 4.7. Hardness

The hardness measurements showed, that the local inhomogeneities has an influence on the mechanical properties. Local minima and softer areas (Figure 11) can be attributed to the smallest pores or soft areas of pure Fe and retained austenite. The fact that the mean hardness value of M1* (Table 7) is higher than the other could be caused by the higher Si and C content (Table 5) and lower resulting retained austenite content (Figure 10). However, these increase the wear properties in use, which could be interesting for various applications. This demonstrates that the LPBF parameters, the powder blend design and used raw alloys can adjust the subsequent parameters of the material properties.

## 5. Conclusions

A Fe-based model alloy ($Fe_{3.5}Si_{1.5}C$) was chosen to demonstrate the effect of different blending strategies of the raw alloys on the homogeneity of the microstructure after LPBF processing. Following raw alloy powders were used to generate the different blends: Fe, SiC, FeC, FeSi and FeSiC.

- All powder blends were processed to dense and crack- free samples (99.8%) with LPBF by suitable parameters.
- Although the melting temperature of SiC is more than 1000 K higher than the melting temperature of the base element Fe, SiC particles could be completely dissolved. Thus, the alloying elements Si and C could be distributed homogeneously in the melt pool.
- However, blending with the milled SiC particles increased the standard deviation of Si distribution compared to the use of lower melting FeSi. No significant differences in the distribution of the elements Fe, Si, C could be detected using the ternary Fe-Si-C or binary FeSi/FeC as raw alloys. Both blending strategies lead to the same chemical homogeneity.

- Most of the inhomogeneity can be attributed to the incompletely dissolved Fe particles. Presumably, the $d_{90}$ of 78 μm is set too high to guarantee the melting of such large particles. Around these Fe particles, there are areas of lower Si and C content, but also Si-rich regions, resulting in variations of the local microstructure, phase distribution and hardness.
- A full chemical homogeneity was not achieved by LPBF processing of the powder blends. But depending on the aimed application, the reached elemental distribution can be sufficient. These applications include multi material structures such as metal matrix composites to improve local functional properties.
- Incomplete dissolution of raw alloy particles leads to local changes in the chemical composition. Thus, microstructure and properties of the material tend to be impacted locally.
- The particle size distribution of the admixed raw alloy powders affects the dissolution behavior of the particles. To support the dissolution of particles and increase the chemical homogeneity, particles sizes smaller than 60 μm are suggested.
- Fundamentally, the homogeneity of the microstructure and hardness depends on the chemical homogeneity reached during the laser melting process. Since the elements Si and C strongly influence martensite formation (influence on $M_s$) as well as isothermal bainite formation (kinetics), the microstructure and the mechanical properties only reflect the local chemical composition. The three blends show different retained austenite contents of 20 to 37 wt. %. Based on the volume and hardness of the retained austenite (150–200 HV1) and empirical values, it can be assumed that up to 20 vol. % retained austenite exert merely slight influence on the overall hardness. Therefore, the overall somewhat lower average hardness of blends M2 and M3 tends to be caused by the larger proportion of stabilized retained austenite.

**Author Contributions:** Conceptualization: A.L.S., V.U. and A.R.; Data curation: M.S.; Formal analysis: A.L.S., M.S. and F.G.; Funding acquisition: V.U., F.W. and R.F.-H.; Investigation: A.L.S.; Methodology: A.L.S., V.U. and M.S.; Project administration: V.U., M.S., A.R., F.W. and R.F.-H.; Resources: V.U., M.S. and R.F.-H.; Supervision: V.U. and M.S.; Validation: A.L.S., V.U. and M.S.; Visualization: A.L.S.; Writing—original draft: A.L.S., V.U. and M.S.; Writing—review & editing: F.G. and A.B.C. All authors have read and agreed to the published version of the manuscript.

**Funding:** The authors gratefully acknowledge the funding by the German Research Foundation (DFG) of project 409651875 within the priority program (SPP) 2122 "Materials for Additive Manufacturing (MATframe)".

**Institutional Review Board Statement:** Not applicable.

**Informed Consent Statement:** Not applicable.

**Data Availability Statement:** Data sharing is not applicable to this article.

**Conflicts of Interest:** The authors declare no conflict of interest.

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
