# Peer review of "Comparison of the Processability and Influence on the Microstructure of Different Starting Powder Blends for Laser Powder Bed Fusion of a Fe3.5Si1.5C Alloy"

_metals, doi:10.3390/met11071107_

Round 1
Reviewer 1 Report
General
1 – Language should be improved, and proof-reading by a native speaker to English editing service is required. In the present form, there are many issues with wording, repeating and construction of sentences. The language check is an obligatory requirement.
2 – Authors present section Results but there is no section Discussion. This is confusing. In section 3 Results there is text that can be considered as discussion, but sometimes discussion comes before presentation of results. Please check carefully text structure and differentiate results and discussion in a logical manner, first – results, then relevant discussion.
Title
-In the present form title does not reflect content of the article. Comparison of what? Samples of what composition? Microstructure, phase composition, alloy composition, powder blend keywords have to be used in title. Powder blending strategies is a confusing term. Authors used 3 different precursor blends, but what means blending strategy is unclear.
Introduction
-In first paragraph tooling applications are mentioned. It is nevertheless unclear what is the relevance between the investigated material, and mentioned tooling applications.
-In principle Introdiction is about steels, but a relevance of that text to the investigated model alloy is unclear. Please explain selection of the model alloy, and show relevance to the text about steels more explicitly.
-Aims are only about homogeneity, but microstructure and phase composition were also intensively investigated. Why they are missed in Aims?
Materials and method
Table 3, mass% or weight% used in the manuscript? In table 3 - mass%, but in table 4 - wt%.
Fig 5 – move composition of etchant to Methods instead of figure caption.
Results
Section 3.3. It would be good to have images where shape and depth of molten pools are presented for all three materials. It could support possible discussion about an influence of the precursor on Marangoni flows and convection in liquid.
Section 3.4. “linear pore pattern, which can be explained by the turning points” – please explain and discuss I the manuscript this mechanism of the pore formation.
Section 3.5. “In addition, the specific effect of Si, which suppresses the carbide precipitation, makes the Si distribution in the alloy traceable from the microstructure formation.” – please explain in the manuscript more clearly how, and how this effect was used in the article.
Section 3.5. Text in lines 326-340 looks like an explanation to the results presented later in lines 340-360.
Section 3.5. Text in lines 350-360 is confusing. Bases on SE image authors conclude presence of retained austenite and unmoltem Fe particles, but how is it possible to conclude chemical composition and phase based on Fig 5? It seems authors brought up here results from investigations presented by further EDX and XRD observations. Authors should indicate in Figure 5 areas with special microstructures, but not conclude what these areas are before presenting more data on chemical composition and phase constitution.
Section 3.6. Figure 6 – what is ESD? In Methods you used EDX.
Section 3.6. “predominantly bainitic microstructure is formed owing to the locally decreased C and Si”. Do these areas formed at solidification? Was accumulated temperature and thermal cycles not enough to homogenize the material? Please elaborate discussion about that.
Section 3.6. “One can observe an internal seam in the particles due to C diffusing into the Fe particle during the LPBF process, increasing the local C content.” From figure 6 this conclusion can not be done. Bright ring around particles can be a charging effect at the locally etched sharp edge. Nevertheless, this is illustrated in Figure 7. Again discussion and conclusions come before results.
Figure 7. Why orange line does not cross the unmolten particle?
Figure 7. There is an area rich of Si in the image, why it was not discussed?
Section 3.7. Please provide discussed Ms temperatures in numbers here as well.
Section 3.7. Austenite phase observed in steels could also be a results of portioning of C in the as-built material due to accumulation of the heat at printing. For example it was observed by Raabe in conventional (http://dx.doi.org/10.1016/j.actamat.2012.01.045) and by Krakhmalev in LPBF steels (http://dx.doi.org/10.1016/j.matdes.2015.08.045). Could the same effect be responsible for high austenite content observed in this investigation? If not, why?
Section 3.7. Add indexes and phases in Figure 9.
Section 3.8. Figure 10 illustrates hardness surfaces rather than curves.
Section 3.8. Are there any reference values for properties of the target alloy? It would be worth to present them. If not, again, why then this material was selected as a target?
Conclusions.
“Most of the inhomogeneity can be attributed to the incompletely dissolved Fe…” – in principle, authors showed more types, (i) – unmolten Fe particles, (ii) areas around them with lower Si and C, (iii) Si-rich regions (Fig 7). All these types have to be mentioned in conclusions.
In principle, authors have to conclude that full homogeneity was not achieved.
Author Response
Please see the attachement

Reviewer 2 Report
It is a correct technological work that examines different strategies of powder blending for alloy development for the laser powder bed fusion, LBDP, process. It is a work linked to new strategies for additive manufacturing from powders. Front my point of view, the scientific soundness is not high. If the editor considers it enough for publication, I suggest to accept after major revision.
XRD analysis has been performed by Rietveld refinement. Thus, information about the quality of the refinement should be given. One option is to provide a refinement parameter, as GOF (goodness of fit).
Likewise, so many crystallographic parameters are also obtained from the Rietveld refinement: crystalline size, microstrain, lattice parameters. This information (with the corresponding errors) should be given in a table.
Some micrographs and compositional analysis show microstructural inhomogeneities. It can be a key factor for applications because functional properties usually are correlated with the microstructure. The authors need to improve related discussion.
Volumetric energy density (VED): Which error is associated to this parameter?
Table 2: Only (wt. %), without numbers in.
Table 5: Errors should be given with two significative figures/digits maximum.
The authors should check the references format taking into account the guidelines of this journal (as an example, reference 29).
I also recommend English revision.
Author Response
We would like to express our gratitude to the reviewers for dedicating their time to evaluating our study. As additional information: Frank Walther is also added as author. The necessary forms have already been accepted at the journal.
We have scrutinized their comments and integrate their suggestions as detailed below:
Feedback Review 2:
- It is a correct technological work that examines different strategies of powder blending for alloy development for the laser powder bed fusion, LBDP, process. It is a work linked to new strategies for additive manufacturing from powders. Front my point of view, the scientific soundness is not high. If the editor considers it enough for publication, I suggest to accept after major revision.
Thank you for your comment. After the revision in the presentation of the problem, the structure and discussion we were able to increase the scientific soundness.
- XRD analysis has been performed by Rietveld refinement. Thus, information about the quality of the refinement should be given. One option is to provide a refinement parameter, as GOF (goodness of fit).
Information on the used Rietveld refinement has been added to paragraph 3.7 in the text and through Table 6. There is also added the GOF and Rwp as information about the quality.
- Likewise, so many crystallographic parameters are also obtained from the Rietveld refinement: crystalline size, microstrain, lattice parameters. This information (with the corresponding errors) should be given in a table.
As described in 2. information about the Rietveld refinement are added in chapter 3.7. The added Table 6 contains the data on the lattice parameters and the corresponding errors. Information on crystalline size and microstrain has been added in the methods section of the phase analysis (2.4 Phase analysis).
- Some micrographs and compositional analysis show microstructural inhomogeneities. It can be a key factor for applications because functional properties usually are correlated with the microstructure. The authors need to improve related discussion.
A discussion about the influence of these microstructural inhomogeneities has been added in chapter 4.6. There, a new paragraph has been added to discuss and explain this topic.
- Volumetric energy density (VED): Which error is associated to this parameter?
A paragraph has been added under the introduction of the VED (Chapter 2.3), which describes the errors and inaccuracies of the VED in more detail.
- Table 2: Only (wt. %), without numbers in.
The table 2 caption has been adjusted.
- Table 5: Errors should be given with two significate figures/digits maximum.
The digits of the numbers in Table 5 has been changed to one decimal place.
- The authors should check the references format taking into account the guidelines of this journal (as an example, reference 29).
The references have been changed using the EndNote style provided by the journal.
- I also recommend English revision.
Thank you for this feedback. This has already been agreed with the journal, we will have the paper edited by the English editing service of the journal after acceptance.
Reviewer 3 Report
General comments
In this research, different strategies of powder blending for laser powder bed fusion (LPBF) were examined by using ferroalloy and carbide alloy powders to produce Fe3.5Si1.5C. Three powder blends (M1, M2, and M3) were used in order to produce: (1) Fe powder admixed with SiC, (2) ternary raw alloy FeSiC and (3) FeSi and FeC. Analysis of the microstructures and the mechanical properties of the LPBF samples was performed by using SEM-EDS, WDS and Vickers hardness testing. The obtained microstructure of M1 was inhomogeneous and contained of bainite, martensite, ferrite and retained austenite. The use of the FeSi and FeC and the ternary ferroalloy FeSiC leads to an improved chemical homogeneity of samples produced by LPBF. Nevertheless, the particle size of the used powder played a critical role for the dissolution behavior in LPBF.
The present study is interesting and novel. The results are well presented and explained and this research is relevant for the readers of Metals. Therefore, I recommend publishing this contribution in Metals, MDPI, after minor revision, as explained below.
Abstract
(1) Page 1, line 28: Please change “Therefore, investigations were carried out by electron microscopy” to “Therefore, investigations were carried out by scanning electron microscopy”
1. Introduction
(2) Page 2, line 56: A full stop mark should be added at the end of the sentence. “… chemical homogeneity On the other hand” should be changed to: “… chemical homogeneity. On the other hand”.
2. Materials and Methods
(3) According to which standards the metallographic specimens were prepared?
(4) In the Abstract part it was mentioned that a Vickers hardness test was performed, however, this detail was not mentioned in the Materials and Methods part. Moreover, in the Materials and Methods part it was mentioned that micro hardness testing was performed. Please add the correct information and be uniform in the definitions.
3. Results
(5) Page 5, line 200: The title should be changed to Results and Discussion.
(6) Table 3, mass % should be changed to wt. % like in Table 2.
(7) Page 8, line 318: “(3.5 mass %) and 1.5 mass % C”, should be changed to wt. %.
(8) Page 9, line 330: change mass % to wt. %.
(9) Page 13, lines 440-444: change wt % to wt. %.
(10) Table 5, change wt% to wt. %.
(11) Table 6: you should write the values ​​with only one digit after the decimal point.
4. Conclusions
(12) Page 16, line 536: > 99,8 density, should be changed to 99.8 %.
(13) Page 16, line 554: “austenite contents of 20 to 37 %” should be changed to wt. %.
Author Response
Abstract
(1) Page 1, line 28: Please change “Therefore, investigations were carried out by electron microscopy” to “Therefore, investigations were carried out by scanning electron microscopy”
Thank you for noticing this inaccuracy/error. The sentence has been changed.
- Introduction
(2) Page 2, line 56: A full stop mark should be added at the end of the sentence. “… chemical homogeneity On the other hand” should be changed to: “… chemical homogeneity. On the other hand”.
We have changed the sentence so this has been changed.
- Materials and Methods
(3) According to which standards the metallographic specimens were prepared?
The specimens are prepared according to the ASTM E407 standard. This information is included into chapter 2.5
(4) In the Abstract part it was mentioned that a Vickers hardness test was performed, however, this detail was not mentioned in the Materials and Methods part. Moreover, in the Materials and Methods part it was mentioned that micro hardness testing was performed. Please add the correct information and be uniform in the definitions.
Thank you for this note. Micro hardness test was used, to the text of the abstract is changed from “… Vickers hardness test” to “micro hardness tests”. Also the chapter 2.6 is changed into “Hardness” instead of “mechanical test”.
- Results
(5) Page 5, line 200: The title should be changed to Results and Discussion.
Due to requests for changes from the other reviewers, the chapter has been split into two parts so that there is now an extra "Discussion" chapter.
(6) Table 3, mass % should be changed to wt. % like in Table 2.
(7) Page 8, line 318: “(3.5 mass %) and 1.5 mass % C”, should be changed to wt. %.
(8) Page 9, line 330: change mass % to wt. %.
(9) Page 13, lines 440-444: change wt % to wt. %.
(10) Table 5, change wt% to wt. %.
(6-10) The hole manuscript was reworked, so that all mass % and wt % are changed into wt. %
(11) Table 6: you should write the values with only one digit after the decimal point.
All numbers are changed to only one digit after the decimal point.
- Conclusions
(12) Page 16, line 536: > 99,8 density, should be changed to 99.8 %.
Done.
(13) Page 16, line 554: “austenite contents of 20 to 37 %” should be changed to wt. %.
Done.
Round 2
Reviewer 1 Report
Two minor comments
1 - Please do not forget the English check.
2 - Authors misinterpreted question about EDS/ESD/EDX. Usually Energy-dispersive X-ray spectroscopy is shortened as EDS or EDX, but not ESD. Authors unified usage of acronym but used ESD, the wrong one.
Reviewer 2 Report
The authors take into account the comments of the referee.
I suggest to accept this improved manuscript.